# Radiomics and Radiogenomics in Differentiating Progression, Pseudoprogression, and Radiation Necrosis in Gliomas

**DOI:** 10.3390/biomedicines13071778

**Published:** 2025-07-21

**Authors:** Sohil Reddy, Tyler Lung, Shashank Muniyappa, Christine Hadley, Benjamin Templeton, Joel Fritz, Daniel Boulter, Keshav Shah, Raj Singh, Simeng Zhu, Jennifer K. Matsui, Joshua D. Palmer

**Affiliations:** 1College of Medicine, The Ohio State University, Columbus, OH 43210, USA; sohil.reddy@osumc.edu (S.R.); tyler.lung@osumc.edu (T.L.); shashank.muniyappa@osumc.edu (S.M.); christine.hadley@osumc.edu (C.H.); benjamin.templeton@osumc.edu (B.T.); 2Department of Radiology, The Ohio State University Wexner Medical Center, Columbus, OH 43210, USA; joel.fritz@osumc.edu (J.F.); daniel.boulter@osumc.edu (D.B.); 3College of Public Health, The Ohio State University, Columbus, OH 43210, USA; shah.1572@osu.edu; 4Department of Radiation Oncology, The Ohio State University Wexner Medical Center, Columbus, OH 43210, USA; raj.singh@osumc.edu (R.S.); simeng.zhu@osumc.edu (S.Z.); 5Department of Radiation Oncology, Stanford Medicine, Stanford, CA 94305, USA; jmatsui@stanford.edu

**Keywords:** glioma, glioblastoma, radiomics, radiogenomics, pseudoprogression, radiation necrosis, tumor progression, tumor recurrence

## Abstract

Over recent decades, significant advancements have been made in the treatment and imaging of gliomas. Conventional imaging techniques, such as MRI and CT, play critical roles in glioma diagnosis and treatment but often fail to distinguish between tumor pseudoprogression (Psp) and radiation necrosis (RN) versus true progression (TP). Emerging fields like radiomics and radiogenomics are addressing these challenges by extracting quantitative features from medical images and correlating them with genomic data, respectively. This article will discuss several studies that show how radiomic features (RFs) can aid in better patient stratification and prognosis. Radiogenomics, particularly in predicting biomarkers such as *MGMT* promoter methylation and *1p/19q* codeletion, shows potential in non-invasive diagnostics. Radiomics also offers tools for predicting tumor recurrence (rBT), essential for treatment management. Further research is needed to standardize these methods and integrate them into clinical practice. This review underscores radiomics and radiogenomics’ potential to revolutionize glioma management, marking a significant shift towards precision neuro-oncology.

## 1. Introduction

Gliomas represent a group of brain tumors that originate from glial cells, which are vital for supporting neurons in the brain. Over the past few decades, significant advancements have been made in the treatment and imaging techniques of gliomas which include surgical procedures, radiation therapy, immunotherapy, and chemotherapeutic treatments [1]. Conventional imaging techniques, such as MRI and CT, have been instrumental in diagnosing and treating gliomas. However, these techniques often struggle to differentiate tumor pseudoprogression (Psp) and radiation necrosis (RN) from true progression (TP) which is crucial for the management and treatment planning for patients [2]. The emerging disciplines of radiomics and radiogenomics offer solutions to these challenges. Radiomics extracts and analyzes a vast array of advanced quantitative features from medical images. Concurrently, radiogenomics correlates these features with genomic data. These methods hold the potential to revolutionize glioma management by offering a non-invasive approach to monitor disease progression and treatment response.

Herein, this review will highlight recent studies utilizing radiomics and radiogenomics in gliomas. We encourage readers to refer to previous reviews that have covered this topic [1,2]. We will discuss how radiomic features (RFs) have been found to correlate with glioma grade, molecular markers, and survival predictions for better patient stratification and prognosis prediction. We will also cover radiogenomics and its potential for predicting biomarkers in GBM such as *MGMT* promoter methylation status and *1p/19q* codeletion status, based on MRI features [3,4]. The *MGMT* promoter methylation status, a key biomarker in GBM, could potentially spare patients the need for invasive tissue sampling and provide real-time information during surgery. The *1p/19q* codeletion status, another essential biomarker, is associated with a favorable prognosis and increased sensitivity to chemotherapy and radiotherapy [5].

Additionally, this article will highlight the role of radiomics in predicting tumor recurrence (rBT). This is crucial as accurate prediction can guide surgical planning and post-operative monitoring. Numerous studies have demonstrated that RFs, such as texture feature analysis, can help identify subtle changes in the tumor microenvironment before they become visible on conventional imaging, thereby providing early signs of rBT location [6]. However, more research is needed to validate these findings and to overcome the current challenges such as the need for standardization, the variability in imaging acquisition protocols, and the integration of these advanced techniques into clinical practice.

To provide a comprehensive and up-to-date review of the role of radiomics and radiogenomics in glioma diagnosis, progression assessment, and recurrence prediction, we conducted a structured literature search across multiple academic databases, including PubMed, Scopus, Web of Science, and Google Scholar. The literature search focused on peer-reviewed articles published from 2005 onward, with an emphasis on studies evaluating radiomics and radiogenomics methodologies in gliomas. The search terms were selected to focus on the role of radiomics and radiogenomics in glioma diagnosis, grading, and recurrence prediction. Keywords included: “radiomics,” “radiogenomics,” “glioma,” “glioblastoma,” “glioblastoma multiforme,” “pseudoprogression,” “true progression,” “radiation necrosis,” “tumor recurrence,” “machine learning,” “deep learning,” “support vector machine,” “convolutional neural networks,” “functional imaging,” “multiparametric MRI,” and “tumor classification.”

Studies were selected based on the following inclusion criteria: original peer-reviewed research articles, studies involving patients with histopathologically confirmed WHO-grade gliomas, including both lower-grade gliomas (LGG) and high-grade gliomas (HGG), research that utilized radiomics-based machine learning models to classify gliomas and/or differentiate tumor recurrence from pseudoprogression and/or predict treatment outcomes, studies integrating functional imaging with radiomics or radiogenomics, and articles that reported performance metrics (e.g., sensitivity, specificity, accuracy) for glioma classification models.

The key exclusion criteria were as follows: studies that did not utilize machine learning or deep learning for glioma classification, articles that did not focus on distinguishing pseudoprogression from true progression or radiation necrosis from tumor recurrence, case reports, and studies with small sample sizes (<30 patients) or lacking sufficient data for performance evaluation.

Following the selection process, 36 studies met the inclusion criteria and were included in the final review. These studies were critically analyzed to highlight advancements in radiomics-driven glioma classification, integration with functional imaging, and machine learning-based approaches for predicting tumor progression and recurrence.

## 2. Background

### 2.1. Gliomas

Gliomas are brain tumors that are categorized based on their histopathological features into four grades according to the World Health Organization (WHO) classification. Grades 1 and 2 are considered LGGs and HGGs are Grades 3 and 4. LGG are slow-growing and can transform into higher-grade gliomas (HGGs). Grade 4 gliomas, also referred to as glioblastoma (GBM), are the most aggressive and lethal form comprising 57% of gliomas, making them the most common brain tumor. Despite advances in treatments, the prognosis of HGGs remains poor with a median survival of 15 months for GBM. This highlights the urgent need for more precise diagnostic and prognostic tools and more effective therapies.

### 2.2. Imaging and Treatment Strategies

The difficulty in treating these tumors stems from a culmination of factors; the primary goal of treatment is complete tumor removal without compromising neurological function, but this is often limited due to gliomas infiltrating surrounding structures. A combination of radiation therapy and chemotherapy is typically administered both before and after surgical intervention [3]. A 2019 review article by Winter et al. [4] emphasized the importance of cancer treatment options and the possible negative outcomes on the central nervous system. They highlighted treatment-induced brain tissue necrosis, often referred to as RN, as a significant obstacle for physicians due to the high morbidity and mortality associated with it. Conventional MRI often falls short of accurately diagnosing this condition and often requires an invasive surgical biopsy. There is an increasing body of literature that features the importance of advanced imaging in RN diagnosis but consensus on an optimal non-invasive imaging method has not been reached due to limitations such as a lack of randomized control studies, poorly matched patient groups, and variability amongst physicians’ preferred treatment methods. However, as we will discuss, numerous studies have indicated that multiple functional imaging modalities can enhance diagnostic accuracy.

Functional imaging techniques have been developed to address some of these challenges by providing detailed physiological and metabolic information about tumor characteristics beyond what standard imaging can reveal. Unlike conventional MRI sequences that assess structural abnormalities, multiple functional imaging focuses on combining various physiological aspects such as tumor metabolism, perfusion, and cellularity. Some of these imaging modalities that have been explored to enhance glioma characterization, including diffusion-weighted imaging (DWI), perfusion-weighted imaging (PWI), and positron emission tomography (PET). DWI quantifies water molecule movement within tissues, allowing differentiation between tumor subtypes based on cellular density; however, its sensitivity to changes in diffusion parameters may lead to confounding results in the presence of inflammation, edema, or hemorrhage, which are typical complications of post-treatment gliomas [5]. PWI measures cerebral blood volume (CBV) to assess tumor vascularity and angiogenesis but can be affected by blood–brain barrier disruption and contrast leakage, leading to inaccurate CBV estimations, particularly in HGGs. PET, particularly with amino acid tracers, has been shown to be more sensitive than conventional MRI in detecting glioma recurrence; nonetheless, PET’s diagnostic specificity can be compromised by non-specific tracer uptake in non-neoplastic conditions such as inflammation or radiation-induced changes, and its limited spatial resolution can impede precise localization [6]. Multiple functional imaging has a decreased risk of misinterpretation over a single imaging modality. The previous gold standard of differentiating treatment outcomes was an invasive biopsy, but Verma et al. [7] proposed an alternative, noninvasive, method of functional imaging to prevent a costly and invasive biopsy from being performed. Previously, functional imagining has been inaccurate, but the use of multiple functional imaging has been recommended because different modalities can provide unique details about tumor physiology. In Table 1, different imaging modalities from studies discussed in this review are presented along with the tumor characteristics of the patients involved in these studies. 

### 2.3. Radiomics and Radiogenomics

In order to address these challenges with managing gliomas, researchers have begun exploring innovative approaches; areas that are growing in popularity are radiomics and radiogenomics which have already shown encouraging results. Radiomics, a rapidly emerging field in medicine, leverages machine learning (ML) to extract a multitude of features from medical images, thereby aiding in clinical decision-making. This high-throughput extraction of quantitative data from radiological images reveals disease characteristics that are often undetectable by even the most experienced clinicians [20]. These features commonly fall into 3 categories: (1) first-order features that describe the distribution of voxel intensities (e.g., mean, standard deviation); (2) shape features that define the geometry of the tumor (e.g., volume, surface area); and (3) texture features, including second-order features, which evaluate spatial relationships between voxel intensities. Techniques like the Gray-Level Co-occurrence Matrix (GLCM) and Gray-Level Run Length Matrix (GLRLM) are employed to quantify tissue heterogeneity [21].

However, one of the primary challenges in radiomics is ensuring reproducibility and consistency of extracted features across different imaging centers and devices. A review of radiomics studies found that inconsistencies stem from variations in image processing, segmentation procedures, and software implementation. For example, discrepancies in segmentation technique such as manual, semi-automatic, and automatic can introduce variability in tumor boundaries, with studies reporting up to 20% variability in texture features between methods [22]. Notably, image processing steps, such as voxel size resampling and denoising, can drastically alter extracted RFs, thereby decreasing the reproducibility. For instance, RFs like entropy and skewness can differ by 10–25% depending on whether isotropic resampling was applied. Isotropic resampling is a preprocessing technique wherein voxel dimensions are interpolated to ensure equal spacing along all three spatial axes, resulting in cubic voxels, which enhances the comparability of RFs across datasets. Furthermore, the stability of shape and texture features are often impacted by both the segmentation approach and the preprocessing pipeline. Features such as sphericity or gray-level co-occurrence matrix (GLCM) contrast have shown intraclass correlation coefficients (ICCs) below 0.6 across datasets using different regions of interest (ROI) definitions, indicating poor reliability. This emphasizes the need for standardization in defining ROIs. Currently, the lack of uniform ROI definitions leads to differences in feature values by as much as 30% depending on whether the entire tumor, enhancing region, or peritumoral edema is segmented. Studies have also highlighted that the cutoff thresholds used to define reproducible features vary widely, ranging from ICC > 0.75 in some studies to >0.9 in others, resulting in inconsistent feature selection. This underscores the necessity for transparent reporting of statistical criteria used in feature selection. Such criteria include *p*-value thresholds for univariate analyses, multicollinearity checks, and validation strategies—yet a 2018 review revealed that fewer than 40% of studies explicitly stated these selection rules, undermining the replicability of findings [22].

The clinical application of radiomics lies in its potential to provide more precise and personalized treatment strategies. This is achieved by integrating complex imaging data with clinical and genomic information, thereby creating a related discipline termed radiogenomics [23]. These techniques offer a more comprehensive picture of the tumor’s characteristics and behavior and may be more accurate than conventional methods used for glioma grading. Currently, glioma grading relies on histopathological analysis of biopsy specimens which is an invasive procedure posing inherent risks to the patient [24]. Moreover, this method provides only a limited view of the tumor, potentially missing other sections due to the heterogeneity of gliomas [25]. In contrast, radiomics has the potential to capture the entire tumor complexity, paving the way for more accurate and holistic glioma grading. In Figure 1, a schematic flowchart is shown illustrating the differences between the use of radiomics and radiogenomics which will be discussed further in the following sections.

According to the WHO guidelines, the classification of a glioma requires specific molecular parameters alongside histological information. Crucial among these molecular parameters are the mutations of the isocitrate dehydrogenase enzyme and arms of chromosomes 1p and 19q (*1p/19q*). The detection of these mutations significantly influences treatment planning. Typically, clinicians collect and analyze tissue samples using immunohistochemistry following surgical tumor removal to detect these mutations. However, recent advancements in artificial intelligence may provide a non-invasive means to assess molecular alterations from MRI images alone. A 2023 article by Chakrabarty et al. introduced a convolutional neural network model capable of classifying IDH mutations and *1p/19q* codeletion status using pre-operative magnetic resonance sequences [26].

## 3. Clinical Application of Radiomics

### 3.1. Identifying Pseudoprogression and True Progression

The ability to predict progression is paramount in glioma management as it directly influences the development of individualized treatment plans. An accurate prediction can guide clinicians in determining therapeutic strategies, the timing of interventions, and the necessity of closer monitoring. Hence, the enhanced predictive power of radiomics not only promises to improve clinical decision-making but also holds the potential to significantly enhance patient outcomes. TP in glioma refers to the actual increase in tumor size or the emergence of new lesions due to disease progression. One study sought to determine if an ML model could more accurately predict the probability of TP better than neuroradiologists. Both the physicians and the ML model were shown three sets of MRIs from three different visits for each patient. After the first set of images, physicians were not able to determine TP (*p* = 0.87) because the patterns seen on the images most likely correlated with Psp. Psp refers to an increase in lesion size or appearance of new lesions caused by treatment-related changes rather than tumor progression [27]. Even at the second and third visits, the physicians’ accuracy ranged from 62 to 72%, while the model had 75% and 81% accuracy at the second and third visits, respectively. The study concluded that first and second-order texture features, which were imperceptible to the neuroradiologists, could be successfully identified by a computer model [28].

### 3.2. Texture Features Analysis and Glioma Grading

In the context of radiomics, texture features refer to quantifiable patterns or structures in the image that are often invisible to the naked eye but are detectable through computational algorithms. They provide a measure of the variations in pixel intensities, or grayscale values within an image and can capture the spatial distributions and relationships reflecting the heterogeneity of the tumor. Specifically, texture features can help in identifying areas of necrosis, edema, and active tumor growth, which can provide valuable insights into the tumor’s behavior and allow for more accurate tumor grading. This is especially significant in the case of gliomas, which are known for their intratumoral heterogeneity, a key factor influencing their aggressiveness and treatment response [1]. Given this significant heterogeneity in morphology and genomic profile, traditional manual segmentation often fails to capture these variations effectively. Studies have shown that deep learning-driven automatic segmentation enhances tumor delineation, ensuring more consistent feature extraction. Additionally, multi-region sampling has been proposed as a method to better represent tumor heterogeneity by extracting radiomic features from multiple regions within the tumor rather than relying on a single volume of interest [29].

Glioma grading is a system used to classify gliomas based on their histological features; it is a key determinant of treatment planning and prognosis but can be challenging due to its heterogeneous nature. However, texture features in radiomics offer a promising avenue for improving glioma grading accuracy. One study sought to verify the increased efficacy of texture features gleaned from multiparametric MRI for glioma grading in comparison to commonly used simple parameters. In this retrospective analysis, a total of 153 patients (42, 33, and 78 patients with Grades 2, 3, and 4 gliomas, respectively) underwent a 3.0T MRI multisequence imaging protocol. Then, high-throughput features were formed from patients’ volumes of interests (VOIs) and adopted to determine the features of LGG vs. HGG, and Grade 3 vs. 4 glioma classification tasks. Accuracy and area under the curve (AUC) were later used to assess grading efficiency and were found to be 96.8%/0.987 for LGGs vs. HGGs, and 98.1%/0.992 for grades 3 vs. 4. The results demonstrated that texture features from multiparametric MRI are superior in glioma grading than simple parameters. Thus, radiomic MRI analysis has the potential to assist in clinical analysis in varied glioma grading [8]. Table 2 below demonstrates several studies and their efficacies in grading gliomas below.

### 3.3. Predicting Tumor Recurrence Location and Enhancing Survival Prediction

In addition to glioma grading, radiomics also has significant potential in predicting the location of glioma recurrence. This application of radiomics is of immense clinical significance as an early prediction of rBT allows clinicians to intervene proactively to guide surgical planning and post-operative monitoring. The application of radiomics allows for the detection of subtle changes in the tumor microenvironment before their visibility on conventional imaging, providing a powerful tool for early recognition of rBT in gliomas.

A 2016 study by Akbari et al. [15] examined GBMs preoperatively using multiparametric MRI in 31 patients and was subsequently cross validated in another cohort of 34 patients. Data from the imaging of these patients’ tumors via ML revealed biomarkers that could better predict where the tumor might likely recur. Multiparametric MRI involves the use of multiple MRI sequences such as T1, T2, FLAIR, and diffusion-weighted imaging to provide a comprehensive view of tumor characteristics including cellularity, edema, and vascularity. This analysis yielded 91% sensitivity and 93% specificity with a mean AUC of 0.84, further suggesting the texture features in these images could predict rBT and better direct therapy in these patients. The main objective of this study was to create a basis of evidence for the existence of such biomarkers and these results create an incentive for further research in determining how these findings can influence predicting rBT.

Glioma recurrence typically occurs at or near the site of the original tumor, often within the resection margins or the peritumoral region [32]. Understanding these patterns of rBT is crucial for guiding clinical decisions; however, predicting the exact location of rBTs remains a challenge. This is where radiomics may offer a potential solution.

One study conducted by Ren et al. in 2023 [33] developed machine learning models based on radiomic analysis of post-operative gliomas to distinguish tumor recurrence from treatment-related effects. The researchers developed several models that accurately distinguished between rBT and treatment-related effects, such as the support vector machine (SVM) model and the k-Nearest Neighbor (KNN) model. The SVM model demonstrated superior diagnostic efficacy across nearly all modalities examined. In a multimodal analysis of both post-operative enhancement regions and edematous regions, the SVM model achieved a 90% accuracy rate compared to an 85% accuracy with the KNN model in differentiating true recurrence from post-treatment changes [34]. Notably, the post-operative enhancement model demonstrated a larger AUC than the edematous model, suggesting that radiomic analysis within edematous areas is not as accurate. Nevertheless, the analysis of the edematous region still achieved an accuracy of 82.5%, indicating that important clinical data can be gleaned from these areas. However, the study also noted the need for more comprehensive datasets, advocating for multicenter collaborations to mitigate bias and enhance the statistical power of the results.

A systematic review in 2020 conducted by Sohn et al. [33] examined the diagnostic accuracy of various ML-based radiomic models and found that the SVM model, with recursive feature elimination (RFE) was superior to 25 other ML-based classifiers in glioma grading. This aligns with prior findings demonstrating that SVM’s ability to handle high-dimensional radiomic data makes it particularly effective in classification tasks involving medical imaging. SVM works by identifying the optimal hyperplane that separates classes with the maximum margin, making it robust against overfitting in smaller datasets, a common issue in radiomics research. Conversely, the KNN model relies on feature similarity, classifying new cases based on their nearest neighbors in the dataset. While KNN is advantageous in non-parametric and highly flexible models, its performance can degrade in large datasets, necessitating feature selection or dimensionality reduction techniques for optimal use [35].

As we transition from the topic of rBT prediction, it is important to note that the potential of radiomics extends far beyond this application. By integrating radiomics with clinical and genomic data, a practice termed radiogenomics, we can significantly improve survival outcomes. One study conducted by Bae et al. [10] examined if RFs from MRI data enhance survival in patients with GBM when utilized alongside genetic and clinical profiles. Multiparametric MRIs of 217 patients who were diagnosed with GBM were retrospectively reviewed to assess for RFs. These features were analyzed using a random survival forest model (RSF) along with genetic and clinical profiles to assess overall survival (OS) and progression-free survival (PFS). Survival was further characterized by utilizing an integrated area under the curve (iAUC). The median PFS was 264 days (range, 21–1809 days) and successfully validated (iAUC: 0.590, 95% CI: 0.502, 0.689, *p* = 0.03). The median OS was 352 days (range, 20–1809 days) and successfully validated (iAUC: 0.652, 95% CI: 0.524, 0.769, *p* = 0.04). Thus, the use of radiomic MRI analysis was found to improve survival prediction when used alongside genetic and clinical profiles compared to models lacking this integration. Therefore, radiomic MRI phenotyping can serve as a potential imaging biomarker and improve survival prediction in patients with GBM.

## 4. Enhanced Diagnostic Accuracy Using Radiomics and Radiogenomics

### 4.1. Differentiating Pseudoprogression from True Progression

One of the applications of radiomics in neuro-oncology is distinguishing Psp from TP in gliomas. This distinction is important as Psp can manifest itself as new clinical symptoms related to the tumor mass effect which may require changes to a patient’s treatment regimen [11]. TP, on the other hand, denotes a real expansion in tumor size or the appearance of new lesions because of advancing disease. The accurate differentiation between Psp and TP is crucial for managing patient care.

The efficacy of radiomics in distinguishing between Psp and TP was demonstrated by Ari and colleagues in their 2022 case study on HGGs. They constructed ML models using 131 patients’ MRIs taken before radio-chemotherapy. The post-treatment histopathological examination confirmed progressive disease in 64 cases and Psp in the remaining 67. A total of 107 RFs, including patient age and cluster shade of MRI, were derived from these MRIs. Subsequently, Generalized Boosted Regression Models, which are composite learning methods that combine the outputs of multiple weak prediction models—typically decision trees—to improve classification accuracy, were utilized to generate a model incorporating the six most correlative features. When tested on an independent validation group, the model demonstrated a mean AUC of 72.87% [70.18%, 76.28%], a sensitivity of 71.75% [62.29%, 75.00%], a specificity of 80.00% [69.23%, 84.62%], and an accuracy of 76.04% [69.90%, 80.00%], with the numbers in brackets denoting 95% confidence intervals. These promising results suggest the potential for this non-invasive Generalized Boosted Regression Models technique to replace the current gold standard of biopsies in differentiating Psp from TP [17].

Another study investigated the use of radiomics in differentiating between TP and Psp in a retrospective cohort. In 2024, Fu and colleagues developed a multiparametric MRI-based radiomics model and retrospectively analyzed T1-enhanced and T2WI/FLAIR MRI images obtained from a sample of 52 glioma patients. 1137 RFs were extracted, which included first-order statistical features, gray-level run length matrices, shape features, gray-level size zone matrices (GLSZM), and gray-level co-occurrence matrices (GLCM). First-order statistical features describe the distribution of voxel intensities within the ROI, such as mean, variance, skewness, and kurtosis, without accounting for spatial relationships. Gray-level run length matrices (GLRLM) quantify the length of consecutive voxels having the same gray-level intensity in a specified direction, offering insight into textural homogeneity and granularity. Shape features describe the geometric properties of the segmented tumor, including volume, surface area, sphericity, and compactness, which are critical for distinguishing irregular tumor morphologies. GLSZM measures the size of connected regions (zones) that share the same gray-level intensity, thereby capturing patterns of spatial uniformity and heterogeneity. GLCM evaluates how often pairs of voxel intensities occur at a specific spatial relationship, enabling quantification of textural patterns such as contrast, correlation, energy, and homogeneity [36]. A feature selection was conducted using the LASSO regression and Select-Kbest methods, which identified the most predictive features. This process yielded 9 and 10 key features, respectively. These features were then utilized within two ML classifiers: the SVM and logistic regression (LR) classifiers, which were used in a model for a training set and testing set that randomly partitioned the dataset in a 7:3 ratio. Three distinct models were evaluated for their efficacy in distinguishing between glioma recurrence and Psp. The SVM classifier demonstrated the most stability of these models, achieving an AUC of 0.96 in the training set, with a sensitivity of 87%, specificity of 94%, and an accuracy of 89% (95% CI: 0.93–1) [18]. KNN was not utilized in this study due to its lower robustness in handling high-dimensional radiomics data and its reliance on distance-based classification, which is sensitive to feature scaling. These findings reinforce prior research that demonstrated SVM as the superior choice in radiomics due to its strengths in high-dimensional feature spaces and generalizability across larger and more diverse datasets. Given that Psp often mimics TP, implementing SVM-based radiomic models into clinical workflows could significantly improve decision-making and serve as an effective non-invasive technique in making this differentiation.

A recent meta-analysis that examined the use of radiomics to differentiate TP from Psp demonstrated a sensitivity of 95.2% and a specificity of 82.4%. Interestingly, this analysis also found that the combined use of ML alongside advanced MRI techniques surpassed the performance of conventional MRI sequences (dOR 6.55, 95% CI 1.29–33.27, *p* = 0.03) [3]. These techniques have the potential to eliminate variability in imaging interpretation by physicians, thereby standardizing the assessment of GBM progression. This can assist neurosurgeons by allowing for more accurate timing of repeat resections, thereby leading to improved surgical outcomes.

### 4.2. Prediction of Pseudoprogression

Another approach was taken to discern the difference between utilizing clinical features as opposed to RFs in pre-radiotherapy MRIs to predict the likelihood of Psp in GBM patients. In this study, Baine et al. [16] assessed clinical features including age, gender, tumor location, extent of tumor resection, and radiation dose. Conversely, the RFs were analyzed through a robust method of 1000 round, 3-fold cross-validation in a sample of 72 patients. The radiomic analysis was conducted using several combinations of features, with the highest AUC of 0.82 achieved by a two-feature model. The features scrutinized in this combination were the minimum intensity and homogeneity of the volume of interest, with the volume of interest being carefully delineated by expert radiologists. The study concluded that RFs alone demonstrated a superior predictive capacity for Psp compared to clinical features. Moreover, it was found that the integration of radiomic and clinical features did not enhance the accuracy of the predictions. Notably, this study did not incorporate post-treatment MRIs into the analysis as the tumor features used in these predictions could potentially be modified by treatment modalities. Post-treatment imaging can reveal treatment-induced physiological changes, such as edema, necrosis, or inflammation, that are often indistinguishable from TP but may manifest with unique radiomic signatures. Incorporating these post-treatment features into predictive models could help capture the evolution of the tumor microenvironment in response to therapy, potentially enhancing model specificity and reducing false positives. Additionally, longitudinal analysis of changes between pre- and post-treatment radiomic features may further improve model robustness and temporal prediction accuracy. The findings underscore the necessity for further research to explore how post-treatment features might contribute to the accurate identification of Psp [16].

### 4.3. Using Magnetic Resonance Contrast Agents

While these studies demonstrate the significant potential of radiomics in differentiating Psp from TP in gliomas, the combination with MR contrast agents can further enhance this capability. Mammadov and colleagues investigated whether the use of contrast agents alongside radiomics could improve the accuracy of distinguishing HGGs from Psp. Their study included 124 patients, 61 of whom had progressed, and 63 had not, all of whom underwent T1-weighted and contrast-enhanced MRIs. This sample was deemed statistically adequate, with a balanced distribution of cases and controls allowing for robust ML model training and validation. Power analysis was not reported in the original study, but the authors referenced prior radiomic studies with comparable cohort sizes, suggesting that this sample size provides sufficient statistical power for initial model development. A Generalized Boosted Regressions Model was developed by sequentially adding the most important features, finding that six features yielded the most accurate model for both contrast-enhanced and non-contrast-enhanced images. The results showed that contrast-enhanced MRIs had significantly higher values compared to non-contrast MRIs: AUC (0.819 [0.760–0.872] vs. 0.651 [0.576–0.761]), mean sensitivity (0.817 [0.750–0.833] vs. 0.616 [0.417–0.833]), mean specificity (0.723 [0.588–0.833] vs. 0.578 [0.417–0.750]), and mean accuracy (0.770 [0.687–0.833] vs. 0.597 [0.500–0.708]). These differences were statistically significant based on the 95% confidence intervals, which showed minimal to no overlap between performance metrics of contrast-enhanced and non-contrast-enhanced models. The higher lower bounds of the confidence intervals for contrast-enhanced metrics compared to the upper bounds for non-contrast metrics further validate the statistical relevance. These findings suggest a new standard of care that uses T1-weighted contrast-enhanced MRIs to more accurately differentiate HGGs from Psp, enabling better predictions and management of HGGs [19].

### 4.4. Radiogenomics and Molecular Markers

Although MR contrast agents in conjunction with radiomics yield promising results, additional factors when combined with radiomics can further improve clinical outcomes of glioma patients. One such feature is *MGMT* promotor methylation status. *MGMT* promoter methylation status refers to an epigenetic modification that occurs on the promoter region of the *O6-methylguanine-DNA methyltransferase* (*MGMT*) gene. This methylation inhibits the expression of *MGMT*, a DNA repair enzyme. In the context of GBM, *MGMT* promoter methylation is typically associated with a better response to alkylating chemotherapy, as the reduced expression of *MGMT* allows these drugs to induce more effective DNA damage in tumor cells, leading to cell death. Therefore, determining *MGMT* promoter methylation status is crucial in guiding therapy decisions and predicting patient prognosis in GBM patients [37]. One study revealed that in patients treated for GBM, Psp was present in 21 (91%) of the 23 patients with hypermethylation and in 11 (41%) of the 27 patients who received treatment with unmethylated *MGMT* promoter (*p* < 0.001). The statistical analysis used to compare these proportions was a chi-square test or Fisher’s exact test (appropriate for small sample sizes), confirming a strong correlation between *MGMT* methylation and an increased incidence of Psp. The 50% difference in Psp occurrence between the two groups (91% vs. 41%) reinforces the clinical relevance of this finding, suggesting that *MGMT* methylation status may be a good predictor of Psp in treated GBM patients. This study showed an association of Psp and *MGMT* methylation with extended median survival [38]. Since Psp typically precedes TP, accurate differentiation of the two becomes crucial as GBM progresses. Furthermore, imaging–genomic correlations hold potential for tracking tumor evolution over time. Longitudinal radiomics analysis has been used to monitor changes in tumor microenvironmental characteristics, which may reflect alterations in genetic expression during treatment progression. Integrating radiomics with molecular data can thus enhance patient-specific treatment strategies and provide a real-time, non-invasive biomarker for glioma management [39].

One of the key challenges in radiogenomics is the high dimensionality of extracted features, which increases the risk of false positive associations. A systematic review found that many radiomics studies suffer from inconsistent validation strategies, leading to inflated model performance and increased false discovery rates. This issue primarily stems from practices such as using the same dataset for both feature selection and model validation (i.e., data leakage), lack of external validation cohorts, and insufficient cross-validation techniques. Without external validation, models may be overfit to specific institutional imaging protocols or patient populations, falsely appearing highly accurate. Moreover, internal validation methods such as k-fold cross-validation, if not properly stratified or repeated, can yield optimistic performance metrics by failing to reflect real-world variability. Studies often lack independent test sets to assess generalizability and fail to report performance variance across folds or resampling iterations, making it difficult to assess model stability. These limitations can significantly inflate reported metrics like AUC, accuracy, or sensitivity, leading to overestimation of clinical utility and a higher likelihood of false discoveries when models are applied to new data [40]. To mitigate these risks, feature selection techniques such as Recursive Feature Elimination (RFE) and LASSO regression have been widely implemented to reduce overfitting and eliminate redundant features. RFE is a wrapper-based feature selection method that recursively removes the least important features based on a model’s performance, typically using a machine learning estimator such as a SVM or random forest. The algorithm evaluates model accuracy after removing each feature, ranks them by importance, and iteratively eliminates the least impactful ones until an optimal subset is identified. This approach ensures that only the most informative features are retained, enhancing model generalizability. LASSO regression is a regularization technique that introduces a penalty term equal to the absolute value of the magnitude of coefficients. By shrinking less important feature coefficients to zero, LASSO effectively performs variable selection and prevents overfitting, especially in high-dimensional datasets with multicollinearity. LASSO is particularly well-suited for radiomics, where the number of features often exceeds the number of samples, by ensuring a sparse and interpretable model [41]. Additionally, external validation with independent datasets has been emphasized as a necessary step to ensure model generalizability beyond a single institutional dataset [39]. Ensuring rigorous study design, including pre-registration of radiomics models and transparent reporting of statistical correction methods, is essential for minimizing false positive discoveries and improving clinical translation of radiogenomics.

A study that explored the potential benefits of combining *MGMT* status with radiomics was conducted by Bani-Sadr and colleagues. In their report, they assessed the efficacy of conventional MRI radiomics for differentiating early progression (EP) from Psp in a cohort of 52 GBM patients post-chemoradiotherapy. The researchers employed a random forest algorithm for classification and analyzed a total of 39 selected RFs either individually or in conjunction with *MGMT* promoter methylation status. Additionally, the study attempted to predict OS and PFS by stratifying patients into high or low-risk categories for rBT, using semi-supervised principal component analysis and survival models. The radiomics model alone yielded an accuracy of 76% (95% CI [54.9–90.6]), a sensitivity of 94.1% (95% CI [71.3–99.8]), and a specificity of 37.5% (95% CI [8.5–75.5]) in differentiating EP from Psp patients within the validation set. However, the combined model of radiomics and *MGMT* promoter status demonstrated improved accuracy, sensitivity, and specificity values of 79.2% (95% CI [59.9–92.9]), 80.0% (95% CI [56.3–94.3]), and 75.0% (95% CI [19.4–99.3]), respectively, in the validation set. Furthermore, the model hinted at the potential utility of RFs for predicting OS and PFS, aligning with current studies that indicate a correlation between increased image heterogeneity and poorer prognoses. Overall, this study highlights the enhanced diagnostic performance achievable through combining radiomics with *MGMT* promoter status. However, considering the lack of an external validation cohort and the restriction to conventional MRI data, future research should aim to validate these findings within larger, prospective cohorts [31].

In another study conducted in 2023, researchers devised a novel two-stage *MGMT* Promoter Methylation Prediction model. This model combined traditional RFs, such as the tumor histograms of oriented gradient derived from MRIs (T1w, T2, and T1Gdmp), with latent variables not directly observed. These features were then integrated using a novel Deep Learning Radiomic Feature Extraction module. The resulting model was trained using k-nearest neighbors and SVM classifiers, employing the publicly available RSNA-MICCAI Brain Tumor Radiogenomic Classification dataset (BRATS-2021) as a basis. This dataset comprised 585 patients, classified according to methylation status, along with a validation set of 87 patients without methylation status. The model’s performance was impressive, achieving an accuracy of 96.84 ± 0.09%, sensitivity of 96.08 ± 0.10%, and specificity of 97.44 ± 0.14%. This innovative approach of integrating genomic and radiomic data has significant potential for future advancements in real-time brain tumor surgeries, aiding in the removal of residual tumor cells, and thereby improving surgical outcomes [42]. This research emphasizes the importance of ongoing studies that aim to equip neurosurgeons with improved treatment options for glioma patients.

Transitioning from *MGMT* status, another significant biomarker in the management of gliomas is the previously mentioned *1p/19q* codeletion status. This genetic alteration, involving the simultaneous loss of chromosome arms 1p and 19q, is a characteristic feature of oligodendroglial tumors, a subtype of gliomas. The resulting codeletion, which has been linked to improved prognosis and increased responsiveness to chemotherapy and radiation therapy, has the potential to be non-invasively identified through the field of radiogenomics [43]. A team led by Kocak explored the use of ML-based MRI texture analysis for classifying lower-grade gliomas (LGG) and predicting *1p/19q* codeletion status. Texture features were retrospectively measured utilizing conventional T2-weighted and contrast-enhanced T1-weighted MRI images from 107 patients with LGG. Subsequent analysis of these images through LIFEx software (version 4.6) led to feature classifications using various ML algorithms. Notably, the predictive performance of the different ML algorithms was statistically significant, (χ2(6) = 26.7, *p* < 0.001). The neural network ML had the highest mean accuracy value (83.8%) and mean AUC (0.86). The results from this study demonstrated that 4/5 of LGG can be accurately classified using ML-based MRI texture, suggesting its potential as a promising tool for predicting LGG *1p/19q* deletion status [14].

In addition to the selection of molecular markers, patient selection plays a crucial role in determining the robustness of radiogenomic models. Studies have demonstrated that the inclusion of heterogeneous patient populations—incorporating different tumor grades, molecular subtypes, and imaging protocols—can introduce variability in radiomic feature extraction, potentially confounding model performance. To improve generalizability, recent radiomics studies have emphasized the importance of selecting well-curated patient cohorts and ensuring adequate representation of clinically relevant subgroups. Multi-center studies with standardized imaging acquisition protocols have been shown to enhance reproducibility, as they allow radiomic models to be trained on diverse datasets, reducing the risk of overfitting to institution-specific imaging characteristics [18]. Ensuring balanced patient selection in radiogenomics research is essential for translating these models into real-world clinical practice.

### 4.5. Accurately Identifying RN

Gliomas are traditionally treated with surgical resection followed by a combination of chemotherapy and radiation therapy. However, a primary adverse effect of this treatment is treatment-induced brain tissue necrosis, often referred to as RN, which is a severe, late-onset complication of radiotherapy, characterized by the death of healthy brain tissue due to radiation damage [1]. This side effect poses significant challenges to physicians due to the high morbidity and mortality associated with it. The accurate identification and management of this condition is crucial as it directly impacts patient prognosis and quality of life. However, the diagnosis of this condition poses several significant challenges. Firstly, RN and TP appear similar on routine follow-up imaging. Determining the correct outcome is a high priority regarding treatment and surgical and post-treatment planning of the condition.

Furthermore, in a 2019 review article, Winter et al. highlighted an increasing body of literature emphasizing the importance of advanced imaging in diagnosing RN [4]. This underscores the clinical importance of integrating these imaging techniques, such as radiomics, into practice. In a comprehensive review article published in 2023, Alizadeh et al. [36] investigated the application of radiomics in the post-treatment follow-up of glioma patients. They reported that radiomics-based models are likely more accurate compared to diagnoses made by radiologists. This conclusion was based on the AUC values exceeding 0.90 for these models in differentiating Psp from TP and RN. Despite these promising results, the authors emphasized the need for larger datasets and more imaging modalities to further validate and refine the advancements currently available in the radiomics literature [35].

### 4.6. Differentiating RN from TP

A study conducted by Park et al. [12] demonstrated the potential of using a combination of one of three different feature selection methods (F-score, LASSO, or Mutual Information) alongside an ML method (k-nearest neighbors, SVM, or AdaBoost) to distinguish between recurrent GBMs and RN. The study involved 127 patients with glioblastoma, who had both conventional MRIs (T1 and T2) and diffusion MRIs (from which the apparent diffusion coefficient was calculated). From these images, 263 RFs were identified and preprocessed, with 18–35 features selected for each model to avoid redundancy. The F-score method ranks features based on their ability to discriminate between two classes by calculating the ratio of variance between classes to variance within classes. Mutual Information measures the mutual dependence between the feature and the target variable, selecting features that provide the most information about class membership. In combination with LASSO, these methods isolate the most meaningful features from high-dimensional data. The researchers then constructed different combinations of machine-learning models. The results indicated that the LASSO feature selection method, classified using an SVM, particularly on diffusion MRIs, demonstrated a mean AUC, accuracy, sensitivity, and specificity of 0.9 [0.84–0.95], 80.5% [0.77–0.84], 78.3% [0.64–0.92], and 82.9% [0.74–0.91], respectively (brackets indicate 95% CI). The goal of this method is to overcome the subjectivity associated with interpreting MRIs by utilizing high throughput second-order features.

In the field of radiomics, second-order features refer to textural features that quantify spatial relationships between pixel or voxel intensities within an ROI. Unlike first-order features, which describe only the distribution of individual voxel intensities, second-order features capture patterns of intensity variation and structural organization in tissue. These features are typically derived from statistical matrices such as the Gray-Level Co-occurrence Matrix (GLCM), Gray-Level Run Length Matrix (GLRLM), and Gray-Level Size Zone Matrix (GLSZM). For example, GLCM-based features measure how often pairs of pixel intensities occur at a specific distance and orientation, enabling assessment of texture attributes like contrast, correlation, energy, and homogeneity [44,45]. These features have a dual purpose: distinguishing between RN and rBT and predicting patients’ survival outcomes. A study by Sadique et al. (2023) [13] aimed to develop an effective model capable of this dual purpose. Of the 158 patients who underwent surgery followed by radiation therapy for gliomas, 15 developed RN, and 143 had rBT. Multiresolution radiomic features (MRF) were extracted from MRIs (T1, T2, T1 contrast, and T2 weighted) and used to construct ML models, such as CatBoost. These models then underwent survival analysis using Copula-based methods and the training set was validated using “repeated random subsampling,” demonstrating an AUC, accuracy, and positive predictive value of 0.89 ± 0.055, 80.88 ± 0.061%, and 0.80 ± 0.064, respectively (95% CI). This method outperformed the Synthetic Minority Over-sampling Technique (SMOTE), which had AUC, accuracy, and positive predictive values of 0.835, 91.00%, and 0.73. The repeated resampling dataset outperformed commonly used Python (version 3.10) methodologies to correct imbalance datasets highlighting the challenges of class imbalance in medical datasets. These findings emphasize the pivotal role of advanced ML techniques in enhancing predictive accuracy, potentially leading to early identification of patients requiring aggressive treatment and subsequently reducing healthcare costs.

## 5. Discussion

This review highlights the potential of radiomics and radiogenomics as transformative tools in the clinical management of gliomas, particularly in differentiating Psp from TP, enhancing recurrence prediction and correlating imaging features with molecular profiles. Several critical issues were highlighted throughout this paper that merit further discussion. First, we addressed the variability and instability inherent in radiomic feature extraction. Discrepancies in segmentation methods and inconsistencies in defining ROIs can yield substantial differences in extracted features, as shown by Traverso et al., with variations up to 30% in some metrics depending on the segmentation technique [22]. Second, we elaborated on the limitations of current validation protocols. Many radiomics studies lack external validation and employ insufficient cross-validation, which can lead to inflated model performance and high false discovery rates—a well-recognized issue that threatens generalizability [39]. To mitigate these risks, we detailed robust feature selection techniques such as LASSO and RFE, which help control overfitting in high-dimensional datasets. Third, functional imaging modalities such as DWI, PWI, and PET, while highly informative, have interpretation limitations due to their sensitivity to treatment-related changes or inflammatory signals, as emphasized in recent studies [5,6]. Fourth, we provided clarification on second-order radiomic features, such as GLCM and GLRLM, which quantify spatial intensity relationships and contribute significantly to model accuracy in predicting both recurrence and survival [46]. Lastly, we discussed the impact of integrating molecular markers, such as *MGMT* promoter methylation, with radiomics, highlighting its statistical significance in predicting Psp and its potential for improving treatment personalization [38]. Overall, this review contributes a cohesive synthesis of emerging methodologies, common drawbacks, and validation practices, offering a roadmap for enhancing the reliability and clinical translation of radiomics and radiogenomics in treating patients with gliomas.

## 6. Conclusions

Despite treatment advances, HGGs have poor prognoses, underscoring the need for enhanced diagnostic tools. This review article highlights the enormous potential of radiomics and radiogenomics and their ability to utilize non-invasive techniques to differentiate Psp and RN vs. TP. By predicting glioma recurrence and progression more accurately, radiomics can assist in radiation therapy planning and early indications of recurrence can be detected by MRI texture features.

Although encouraging results have been found, more research is needed to validate the radiomics and radiogenomics findings and address current challenges, such as standardizing imaging protocols to eventually integrate these techniques into widespread clinical practice. Future research should concentrate on studies carried out at multiple medical institutions globally to improve reliability and minimize bias. Additionally, it is essential to continue developing reliable algorithms and tools that can be used for real-time application during neurosurgeries.

The non-invasive nature of radiomics offers significant advantages, reducing the need for biopsies and facilitating more flexible and dynamic strategies based on comprehensive tumor screening. Ultimately, radiomics and radiogenomics could transform neuro-oncology by making diagnoses more precise, guiding treatment choices, and enhancing prognostic accuracy, leading to better patient care.

## Figures and Tables

**Figure 1 biomedicines-13-01778-f001:**
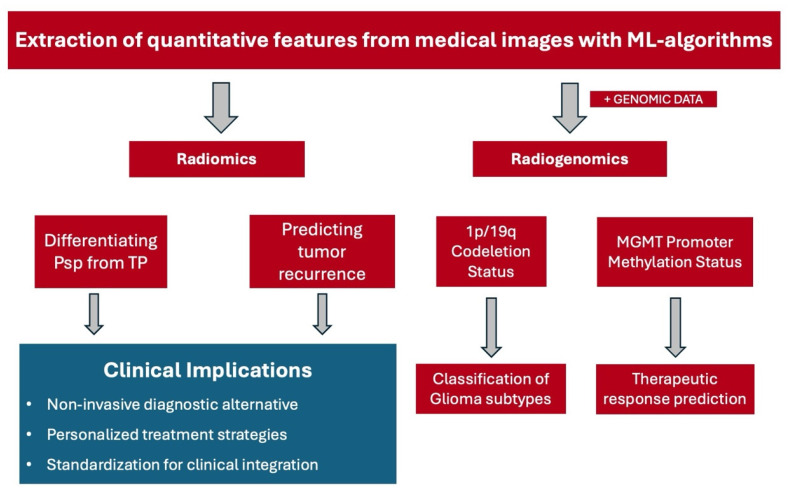
A schematic flowchart that summarizes the major findings and clinical implications of radiomics and radiogenomics in the classification, recurrence, recurrence prediction, and response prediction in glioma management. Abbreviations. ML: machine learning, Psp: pseudoprogression, TP: true progression, *MGMT*: O6-methylguanine-DNA methyltransferase.

**Table 1 biomedicines-13-01778-t001:** Summary of radiomics and radiogenomics studies applied to glioma characterization and prognostication utilized in this review.

Authors	Sample Size	Outcome Predicted	Type of Tumor (n)	ML Methods Used	Imaging Modalities
Turk et al., 2022 [8]	76	Psp vs. TP	GBM (76)	Random Forest, Naïve Bayes	CE-T1WI, T2-WI, ADC
Tian et al., 2018 [9]	153	Survival Stratification	GBM (153)	SVM-RFE	T1, T1-CE, T2, FLAIR
Bae et al., 2018 [10]	248	GBM vs. Brain Metastasis	GBM (159), Metastasis (89)	DNN, AdaBoost	CE-T1WI, T2-FLAIR
Dallabona et al., 2017 [11]	156	Survival Stratification	GBM (156)	LASSO, Logistic Regression	T1-CE, FLAIR, DWI, ADC
Tabassum et al., 2023 [6]	N/A	Method Evaluation	HGG (review)	Summary Review	MRI
Park et al., 2021 [12]	127	TP vs. RN	GBM (127)	SVM (via LASSO, F-score, MI)	T1, T2, ADC
Sadique et al., 2023 [13]	158	Survival prediction and TP vs. RN	GBM (158)	CatBoost, Copula-based survival models	T1, T2, FLAIR, CE-T1
Kocak et al., 2020 [14]	194	Survival Prediction (OS, PFS)	Recurrent GBM (194)	ML classifiers	T2-FLAIR, Gadolinium-enhanced MRI
Akbari et al., 2016 [15]	65	Recurrence location prediction	GBM (65)	N/A	Multiparametric MRI
Baine et al., 2021 [16]	72	Psp prediction	GBM (72)	LR	T1, T2 (pre-radiotherapy)
Ari et al., 2022 [17]	131	Psp vs. TP	HGGs (131)	Gradient Boosting Machines	T1-weighted, T2
Fu et al., 2024 [18]	52	Psp vs. TP	HGGs (52)	SVM, LR	T1-contrast, T2-FLAIR
Mammadov et al., 2022 [19]	124	Psp vs. TP	HGG (124)	Gradient boosting machines	T1-weighted, contrast-enhanced

Abbreviations. FLAIR: fluid-attenuated inversion recovery, ADC: apparent diffuse coefficient, rCBV: relative cerebral blood volume, SVM-RFE: support vector machine-recursive feature elimination, CE: contrast enhanced.

**Table 2 biomedicines-13-01778-t002:** Clinical trials evaluating glioma grading tasks.

Study	Patients	Algorithm(s) Used	Primary Outcome
Tian et al., 2018 [9]	153 (LGG vs. HGG)	NONE	72.5% ACC *; 0.859 AUC
		SMOTE + SVM-RFE *	96.8%ACC; 0.987 AUC
	111 (Grade III vs. Grade IV)	SMOTE +SVM-RFE	98.1% ACC; 0.992 AUC
Zhang et al., 2019 [30]	51 (Glioma Necrosis vs. Recurrence)	Fusion AlexNet	97.8% ACC; 0.9982 AUC
Gutta et al., 2021 [31]		SVM *	56% ACC; 0.43 Precis. *
		RF *	58% ACC; 0.35 Precis.
	237	GB *	64% ACC; 0.40 Precis.
		CNN *	87% ACC; 0.76 Precis.
Bani-Sadr et al., 2019 [32]	53	SCPA *, Random Forest	OS *: HR * 3.63 (Training), HR 3.76 (Validation); PFS *: HR 2.58 (Training), HR 3.58 (Validation)

* Abbreviations. SMOTE: Synthetic Minority Over-sampling Technique, SVM-RFE: support vector machine-based recursive feature elimination, ACC: accuracy, SVM: support vector machine, Precis.: precision, RF: random forest, GB: gradient boosting, CNN: convolutional neural networks, SCPA: supervised principal component analysis, OS: overall survival, HR: hazard ratio, PFS: progression-free survivals.

## Data Availability

No new data were created or analyzed in this study. Data sharing is not applicable to this article.

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
