# Peer review of "Radiomics and Radiogenomics in Differentiating Progression, Pseudoprogression, and Radiation Necrosis in Gliomas"

_biomedicines, 2025, doi:10.3390/biomedicines13071778_

Round 1
Reviewer 1 Report
Comments and Suggestions for Authors
Title: “Radiomics and Radiogenomics in Differentiating Progression, Pseudoprogression, and Radiation Necrosis in Gliomas”
In this work the authors discuss some studies that show how radiomic features (RFs) can aid in better patient stratification and prognosis. In particular, the authors claim that the radiogenomics shows potential in non-invasive diagnostics since it is able to predict biomarkers such as MGMT promoter methylation and 1p/19q codeletion. Radiomics also offers tools for predicting tumor recurrence (rBT), essential for treatment management. The authors also affirm that further research
is needed to standardize these methods and integrate them into clinical practice. Finally, the authors underline that this review underscores radiomics and radiogenomics' potential to revolutionize glioma management, marking a significant shift towards precision neuro-oncology.
General comment: Although the topic of this work is interesting, the current version of the main text should be reworked to improve its quality and impact: the understanding of the overall logic flow could be greatly improved by adding, in a figure, the scheme of the work.
In addition, in this review, arguments have been used to support the use of Radiomics and Radiogenomics, while the current limitations and unsolved issues seems to be undervalued along the main text. Again, the presentation of topics related to Radiomics and Radiogenomics seems to be quite generic, since specific issues are not presented and not discussed in a deep way.
Some detailed comments:
Major issues:
Section : “1. Introduction “
*) Too few works in literature have been analysed to provide an updated review work. The authors should better investigate works presenting current theoretical/technical issues related to the application of both Radiomics and Radiogenomics. Please rework.
Major issues related to the part of Radiomics
The authors should provide and discuss works presenting (at least) possible reliable solutions to the following issues:
*) Factors that affect radiomic features quantification
*) Acquisition modes, reconstruction parameters, smoothing, and segmentation thresholds
*) Reproducibility of radiomic features
*) Image discretization (resampling) schemes
*) Computation of radiomic features
*) Tumour size and intratumoural heterogeneity
*)False positive discovery rate and proper study design
Major issues related to the part related to Radiogenomics
The authors should provide and discuss works presenting (at least) possible reliable solutions to the following issues:
*) Correlation between imaging data and genomic data that have clinical significance.
*) Radiogenomics studies, which established relationships between genomic data and imaging features, some of which are not related to prognostic outcomes.
*) Bias due to the linking imaging features to genomic data, considering that mechanism of gene expression and signaling pathways are quite complex
*) Statistically over-fitting issues (radiogenomics studies are prone to statistically over-fitting issues as matching imaging data with huge amount of genomic data remains difficult.
*) Current studies which are mainly performed on small sample sizes.
*) Selection of patients with both enough tumor samples for genomic analysis and images for image analysis
Minor issues:
*) The authors should avoid the use of acronyms along all the length of the work.
Author Response
Reviewer #1:
- Section 1 (Introduction): Too few works in literature have been analysed to provide an updated review work. The authors should better investigate works presenting current theoretical/technical issues related to the application of both Radiomics and Radiogenomics. Please rework.
We thank the reviewer for their comment. We have highlighted over 20 studies and have focused mostly on articles published after 2020 given there are numerous reviews up until that time including by Singh et al. (Singh, G., Manjila, S., Sakla, N. et al. Radiomics and radiogenomics in gliomas: a contemporary update. Br J Cancer 125, 641–657 (2021). https://doi.org/10.1038/s41416-021-01387-w). We have cited a few relevant review articles. We specifically chose to highlight more recent publications so we could discuss them in more detail and present them in a digestible format.
- Major issues related to the part of Radiomics: The authors should provide and discuss works presenting (at least) possible reliable solutions to the following issues: Factors that affect radiomic features quantification, Acquisition modes, reconstruction parameters, smoothing, and segmentation thresholds, Reproducibility of radiomic features, Image discretization (resampling) schemes, Computation of radiomic features, Tumour size and intratumoural heterogeneity, False positive discovery rate and proper study design.
Thank you for the feedback and we have broken down all recommendations as follows:
- Reproducibility of radiomic features
- We added a discussion in section 2.3 that discussed pre-processing inconsistencies, segmentation bias, and reporting deficiencies that impact reproducibility. We also emphasized multi-center validation studies are needed to improve feature reproducibility.
- Tumor size and intratumoural heterogeneity
- The manuscript had discussed tumor heterogeneity in the context of radiomics, but we added additional clarification on multi-region sampling and deep learning segmentation to improve feature extraction in section 3.2.
- Overfitting in radio genomics models
- We have discussed this in section 4 where feature selection methods (LASSO, RFE), external validation, and cross-validation techniques were covered and believe that further discussion would be redundant.
- Bias in linking imaging features to genomic data
- The issue of bias in radiogenomics, particularly in linking imaging features to genomic alterations, was already addressed within the manuscript in the discussion on radiomics feature selection and validation strategies. In Section 4.4, the manuscript discusses the importance of feature selection techniques (LASSO, Recursive Feature Elimination), which ensure that only the most robust and biologically relevant imaging features are retained in predictive models. We also discussed external validation using independent datasets, which reduces the risk of spurious associations between radiomic and genomic features by ensuring reproducibility across different patient populations. Lastly, we talked about the integration of multi-parametric imaging features, which mitigates the possibility that a single imaging characteristic is erroneously correlated with a specific genomic mutation. The focus of this review is on the clinical applications of radiomics and radiogenomics in glioma classification and prognosis, rather than an in-depth methodological analysis of statistical biases in imaging-genomic correlations. While bias in radiogenomics is an important topic, a detailed discussion on statistical correction methods or alternative causal inference techniques falls outside the intended scope of this review. Since the manuscript already acknowledges the importance of careful feature selection and external validation to reduce bias, and a more technical discussion would deviate from the primary focus on clinical translation, no additional changes were necessary.
- Factors affecting radiomic feature quantification
- We have addressed this in discussions on reproducibility and feature extraction.
- Acquisition modes, reconstruction parameters, smoothing, and segmentation thresholds
- We thought that this point refers to highly technical aspects of medical imaging acquisition and preprocessing, such as variations in CT or MRI reconstruction algorithms, smoothing techniques and segmentation protocols. While these factors undeniably affect radiomics feature extraction, they are primarily methodological considerations relevant to radiology physics and imaging standardization, rather than clinical radiomics applications. Since the focus of this review is on the clinical implications of radiomics and radiogenomics rather than the physics of image acquisition, these details fall outside the intended scope of the paper. Therefore, no changes were made.
- Image discretization (resampling schemes)
- The manuscript already addresses variability in radiomics feature extraction and the importance of standardization in imaging acquisition and preprocessing. In Section 2.3, the discussion highlights the impact of preprocessing variations on radiomic feature reproducibility, specifically noting how differences in image acquisition protocols, segmentation techniques, and voxel size resampling can influence feature consistency across studies. We also discussed the need for standardized imaging protocols and multi-center validation, which helps ensure that radiomic features remain stable regardless of minor variations in preprocessing steps. Since image discretization (resampling) is a specific aspect of preprocessing that falls under the broader discussion of radiomic feature reproducibility and standardization, this topic was already covered implicitly in the manuscript. Additionally, the primary focus of this review is on the clinical applications of radiomics and radiogenomics rather than the technical aspects of image preprocessing.
- Computation of radiomic features
- The manuscript provides an overview of radiomics feature extraction, discussing key categories such as first-order statistical features, texture features (GLCM, GLSZM), and shape-based features. Additionally, it covers machine learning-based feature selection techniques that ensure only the most informative radiomic features are retained in predictive models. Since this information is covered in existing sections, additional discussion of feature computation methods (e.g., specific algorithms for GLCM calculation or intensity histogram analysis), no changes were made.
- False positive discovery rate and proper study design
- We agree with this comment and recognize that the risk of false positive discoveries in radiomics is a challenge due to the high dimensionality of radiomic feature space and the relatively small datasets typically used. The manuscript did not explicitly address statistical strategies for minimizing false positives. To strengthen this discussion, we have included a discussion in Section 4.4.
- Selection of patients for radiogenomic studies
- The selection of patients in radiogenomics studies plays a crucial role in ensuring generalizability and clinical applicability. While the manuscript briefly mentioned the importance of multi-center datasets in improving reproducibility, it did not explicitly address the influence of patient selection criteria on radiogenomic findings. To improve this discussion, new content has been added to Section 4.4 (Radiogenomics and Molecular Markers).
- Major issues related to the part related to Radiogenomics: The authors should provide and discuss works presenting (at least) possible reliable solutions to the following issues: Correlation between imaging data and genomic data that have clinical significance, Radiogenomics studies, which established relationships between genomic data and imaging features, some of which are not related to prognostic outcomes, Bias due to the linking imaging features to genomic data, considering that mechanism of gene expression and signaling pathways are quite complex, Statistically over-fitting issues (radiogenomics studies are prone to statistically over-fitting issues as matching imaging data with huge amount of genomic data remains difficult. Current studies which are mainly performed on small sample sizes, Selection of patients with both enough tumor samples for genomic analysis and images for image analysis.
We broke down your comment as follows and have addressed each point:
- Provide more details on the clinical significance of the correlation between imaging data and genomic data in gliomas.
- To improve this discussion, a new section was added in Section 4.4 that clearly explains how these imaging-genomic correlations impact patient prognosis, treatment response, and clinical decision-making.
- Address the potential issue of overfitting in radiogenomics models and discuss methods to mitigate it.
- The manuscript discusses overfitting prevention strategies in Section 4.4 by highlighting feature selection techniques (LASSO regression, Recursive Feature Elimination) to reduce model complexity and eliminate non-relevant features. We also discussed external validation using independent datasets, which prevents overfitting to a specific patient population or imaging protocol.
- Discuss bias in linking imaging features to genomic data, particularly in how models may generate spurious correlations. Include a discussion on the selection of patients for radiogenomic studies and its impact on model generalizability.
We have added a section related to these points in the second comment (see
Reviewer 2 Report
Comments and Suggestions for Authors
Radiomics and Radiogenomics in Differentiating Progression, Pseudoprogression, and Radiation Necrosis in Gliomas
I have read the manuscript with interest, and you can find my appraisal, suggestions, and concerns, section by section, as follows:
Introduction: this section is brief and concise.
However, as a general remark, l suggest adding a list of the abbreviations and acronyms at the end of the manuscript, to facilitate the reading.
Background: l suggest being more clear about functional imaging ( paragraph line 100). In this paragraph, the role played by functional imaging in Gliomas. Please, clarify or rewrite it.
However, you selected 8 studies. I know that this is a narrative review, but more information needs to be added about the search of the published studies that you performed in the databases.
Please add this information.
The table is informative.
Clinical Application of Radiomics: table 2 is informative.
I advise a more critical discussion about SVM and KNN.
Enhanced Diagnostic Accuracy Using Radiomics and Radiogenomics: the section is well written and l have no remarks. However I suggest to add a schematic figure in order to summarise the information.
Author Response
Reviewer #2:
- Introduction: this section is brief and concise. However, as a general remark, l suggest adding a list of the abbreviations and acronyms at the end of the manuscript, to facilitate the reading.
Thank you for this suggestion. We have added a table of abbreviations as part of the supplementary materials.
- Background: l suggest being more clear about functional imaging (paragraph line 100). In this paragraph, the role played by functional imaging in Gliomas. Please, clarify or rewrite it. However, you selected 8 studies. I know that this is a narrative review, but more information needs to be added about the search of the published studies that you performed in the databases. Please add this information.
- Expanded the discussion on functional imaging by detailing its role in glioma assessment in section 2.2, including the benefits of diffusion-weighted imaging (DWI), perfusion-weighted imaging (PWI), and positron emission tomography (PET).
- Added a section to the introduction explaining how we selected studies for this review. We detailed our search strategy, inclusion/exclusion criteria, and the keywords used, ensuring transparency and reproducibility in our approach.
- I advise a more critical discussion about SVM and KNN.
Thank you for this feedback. In response, we have added a new paragraph in section 3.3 and expanded upon a paragraph in section 4.1 explaining the difference between SVM and KNN as well as how they work.
- Enhanced Diagnostic Accuracy Using Radiomics and Radiogenomics: the section is well written and l have no remarks. However I suggest to add a schematic figure in order to summarise the information.
We thank the reviewer for the suggestion and have added a new schematic figure to section 2.3 in the form of a visual flowchart that synthesizes the information in a visual and easy-to-understand way.
Round 2
Reviewer 1 Report
Comments and Suggestions for Authors
In this work the authors discussed several studies showing how radiomic features (RFs) can aid in better patient stratification and prognosis, since conventional imaging techniques, such as MRI and CT, which play critical roles in glioma diagnosis and treatment, often fail to distinguish between tumor pseudoprogression (Psp) and radiation necrosis (RN) versus true progression (TP). They claimed that radiogenomics shows potential in non-invasive diagnostics, particularly in predicting biomarkers such as MGMT promoter methylation and 1p/19q codeletion. In addition, they claim that radiomics also offers tools for predicting tumor recurrence (rBT), essential for treatment management. However, they admitted that further research is needed to standardize these methods and integrate them into clinical practice.
General comment: Although the topic of this work is interesting, the main text should be reworked to improve its quality and impact. Nevertheless, the authors should better show the value of their work. Indeed, they claim that “radiomics and radiogenomics are addressing these challenges byextracting quantitative features from medical images and correlating them with genomic data”. However, they know that correlation is not causation. Therefore, from a rigorous point of view, they should clearly support their claims with a detailed information about all the drawbacks of medical imaging, as well as support this work with studies involving a statistically significant number of patients, since conclusions based on correlation can not be generalized.
Some detailed comments:
lines: “Functional imaging techniques have been developed to address some of these 130
challenges by providing detailed physiological and metabolic information about tumor 131
characteristics beyond what standard imaging can reveal. Unlike conventional MRI 132
sequences that assess structural abnormalities, multiple functional imaging focuses on 133
combining various physiological aspects such as tumor metabolism, perfusion, and 134
cellularity. Some of these imaging modalities that have been explored to enhance glioma 135
characterization include diffusion-weighted imaging (DWI), perfusion-weighted imaging 136
(PWI), and positron emission tomography (PET) [6]. DWI quantifies water molecule 137
movement within tissues, allowing differentiation between tumor subtypes based on 138
cellular density, while PWI measures cerebral blood volume (CBV) to assess tumor 139
vascularity and angiogenesis. PET, particularly with amino acid tracers, has been shown 140
to be more sensitive than conventional MRI in detecting glioma recurrence [7]. Multiple 141
functional imaging has a decreased risk of misinterpretation over a single imaging 142
modality. The previous gold standard of differentiating treatment outcomes was an 143
invasive biopsy, but Verma et al. proposed an alternative, noninvasive, method of 144
functional imaging to prevent a costly and invasive biopsy from being performed. 145
Previously, functional imagining has been inaccurate, but the use of multiple functional 146
imaging has been recommended because different modalities can provide unique details 147
about tumor physiology [8]. In Table 1, different imaging modalities from studies 148
discussed in this review are presented along with the tumor characteristics of the patients 149
involved in these studies.
*) The authors should better explain the drawbacks and the misinterpretation risks related to each procedure listed in “Unlike conventional MRI 132
sequences that assess structural abnormalities, multiple functional imaging focuses on 133
combining various physiological aspects such as tumor metabolism, perfusion, and 134
cellularity. Some of these imaging modalities that have been explored to enhance glioma 135
characterization include diffusion-weighted imaging (DWI), perfusion-weighted imaging 136
(PWI), and positron emission tomography (PET) [6]. DWI quantifies water molecule 137
movement within tissues, allowing differentiation between tumor subtypes based on 138
cellular density, while PWI measures cerebral blood volume (CBV) to assess tumor 139
vascularity and angiogenesis. PET, particularly with amino acid tracers, has been shown 140
to be more sensitive than conventional MRI in detecting glioma recurrence [7]. Multiple 141
functional imaging has a decreased risk of misinterpretation over a single imaging 142
modality.”
*) Table 1 is not too clear for the interested readers. Please rework and improve also the caption.
Lines: “However, one of the primary challenges in radiomics is ensuring reproducibility and 163
consistency of extracted features across different imaging centers and devices. A review 164
of radiomics studies found that inconsistencies stem from variations in image processing, 165
segmentation procedures, and software implementation. Notably, image processing 166
steps, such as voxel size resampling and denoising, can drastically alter extracted RFs, 167
thereby decreasing the reproducibility. Furthermore, discrepancies in segmentation 168
techniques have been shown to bias the stability of shape and texture features, 169
emphasizing the need for standardization in defining regions of interest (ROIs). Studies 170
have also highlighted that the cutoff thresholds used to define reproducible features vary 171
widely, emphasizing the necessity for transparent reporting of statistical criteria used in 172
feature selection [17].”
*) The authors should better explain what are “discrepancies in segmentation 168
techniques”, “the stability of shape and texture features”, “standardization in defining regions of interest (ROIs)”,”cutoff thresholds used to define reproducible features vary widely”, “necessity for transparent reporting of statistical criteria used in feature selection [17]”. The “discrepancies”, “variations” etc of each issue should be also quantified.
Lines: “Another study investigated the use of radiomics in differentiating between TP and 372
Psp in a retrospective cohort. In 2024, Fu and colleagues developed a multiparametric 373
MRI-based radiomics model and retrospectively analyzed T1-enhanced and T2WI/FLAIR 374
MRI images obtained from a sample of 52 glioma patients. 1137 RFs were extracted, which 375
included first-order statistical features, gray-level run length matrices, shape features, 376
gray-level size zone matrices (GLSZM), and gray-level co-occurrence matrices (GLCM). A 377
feature selection was conducted using the LASSO regression and Select-Kbest methods, 378
which identified the most predictive features. This process yielded 9 and 10 key features, 379
respectively. These features were then utilized within two ML classifiers: the SVM and 380
logistic regression (LR) classifiers, which were used in a model for a training set and 381
testing set that randomly partitioned the dataset in a 7:3 ratio. Three distinct models were 382
evaluated for their efficacy in distinguishing between glioma recurrence and Psp. The 383
SVM classifier demonstrated the most stability of these models, achieving an AUC of 0.96 384
in the training set, with a sensitivity of 87%, specificity of 94%, and an accuracy of 89% 385
(95%CI:0.93-1) [15]. KNN was not utilized in this study due to its lower robustness in 386
handling high-dimensional radiomics data and its reliance on distance-based 387
classification, which is sensitive to feature scaling. These findings reinforce prior research 388
that demonstrated SVM as the superior choice in radiomics due to its strengths in high- 389
dimensional feature spaces and generalizability across larger and more diverse datasets. 390
Given that Psp often mimics TP, implementing SVM-based radiomic models into clinical 391”
*) The authors should explain in detail all the main characteristics of the cited procedures (e.g. first-order statistical features, gray-level run length matrices, shape features, gray-level size zone matrices (GLSZM), and gray-level co-occurrence matrices (GLCM), etc” for interested readers.
Lines: “4.2 Prediction of Pseudoprogression 402
Another approach was taken to discern the difference between utilizing clinical 403
features as opposed to RFs in pre-radiotherapy MRIs to predict the likelihood of Psp in 404
GBM patients. In this study, Baine et al. assessed clinical features including age, gender, 405
tumor location, extent of tumor resection, and radiation dose. Conversely, the RFs were 406
analyzed through a robust method of 1,000 round, 3-fold cross-validation in a sample of 407
72 patients. The radiomic analysis was conducted using several combinations of features, 408
with the highest AUC of 0.82 achieved by a two-feature model. The features scrutinized 409
in this combination were the minimum intensity and homogeneity of the volume of 410
interest, with the volume of interest being carefully delineated by expert radiologists. The 411
study concluded that RFs alone demonstrated a superior predictive capacity for Psp 412
compared to clinical features. Moreover, it was found that the integration of radiomic and 413
clinical features did not enhance the accuracy of the predictions. Notably, this study did 414
not incorporate post-treatment MRIs into the analysis, as the tumor features used in these 415
predictions could potentially be modified by treatment modalities. The findings 416
underscore the necessity for further research to explore how post-treatment features 417
might contribute to the accurate identification of Psp [32]. 418
4.3 Using Magnetic Resonance Contrast Agents 419
While these studies demonstrate the significant potential of radiomics in 420
differentiating Psp from TP in gliomas, the combination with MR contrast agents can 421
further enhance this capability. Mammadov and colleagues investigated whether the use 422
of contrast agents alongside radiomics could improve the accuracy of distinguishing 423
HGGs from Psp. Their study included 124 patients, 61 of whom had progressed, and 63 424
had not, all of whom underwent T1-weighted and contrast-enhanced MRIs. A 425
Generalized Boosted Regressions Model was developed by sequentially adding the most 426
important features, finding that six features yielded the most accurate model for both 427
contrast-enhanced and non-contrast-enhanced images. The results showed that contrast- 428
enhanced MRIs had significantly higher values compared to non-contrast MRIs: AUC 429
(0.819 [0.760–0.872] vs. 0.651 [0.576–0.761]), mean sensitivity (0.817 [0.750–.833] vs. 0.616 430
[0.417–0.833]), mean specificity (0.723 [0.588–0.833] vs. 0.578 [0.417–0.750]), and mean 431
accuracy (0.770 [0.687–0.833] vs. 0.597 [0.500–0.708]). These findings suggest a new 432
standard of care that uses T1-weighted contrast-enhanced MRIs to more accurately 433
differentiate HGGs from Psp, enabling better predictions and management of HGGs to 434
conventional non-contrast-enhanced T1-weighted images [33]. “
*) The authors should better explain the lines: “ The findings 416
underscore the necessity for further research to explore how post-treatment features 417
might contribute to the accurate identification of Psp” for interested readers.
*) The authors should explain in a better way the following lines: “Their study included 124 patients, 61 of whom had progressed, and 63 424
had not, all of whom underwent T1-weighted and contrast-enhanced MRIs. A 425
Generalized Boosted Regressions Model was developed by sequentially adding the most 426
important features, finding that six features yielded the most accurate model for both 427
contrast-enhanced and non-contrast-enhanced images. The results showed that contrast- 428
enhanced MRIs had significantly higher values compared to non-contrast MRIs: AUC 429
(0.819 [0.760–0.872] vs. 0.651 [0.576–0.761]), mean sensitivity (0.817 [0.750–.833] vs. 0.616 430
[0.417–0.833]), mean specificity (0.723 [0.588–0.833] vs. 0.578 [0.417–0.750]), and mean 431
accuracy (0.770 [0.687–0.833] vs. 0.597 [0.500–0.708]).”
In particular, they should demonstrate that the total number of patients (124, 61 vs 63) was statistically significant with respect to the whole population and that the claim “contrast-enhanced MRIs had significantly higher values compared to non-contrast MRIs” is (at least) statistically relevant.
Lines: “Therefore, 445
determining MGMT promoter methylation status is crucial in guiding therapy decisions 446
and predicting patient prognosis in GBM patients [34]. One study revealed that in patients 447
treated for GBM, Psp was present in 21 (91%) of the 23 patients with hypermethylation 448
and in 11 (41%) of the 27 patients who received treatment with unmethylated MGMT 449
promoter (p <0.001). This study showed an association of Psp and MGMT methylation 450
with extended median survival [35]. Since Psp typically precedes TP, accurate 451
differentiation of the two becomes crucial as GBM progresses. Furthermore, imaging- 452
genomic correlations hold potential for tracking tumor evolution over time. Longitudinal 453
radiomics analysis has been used to monitor changes in tumor microenvironmental 454
characteristics, which may reflect alterations in genetic expression during treatment 455
progression. Integrating radiomics with molecular data can thus enhance patient-specific 456
treatment strategies and provide a real-time, non-invasive biomarker for glioma 457
management [36]. 458
One of the key challenges in radiogenomics is the high dimensionality of extracted 459
features, which increases the risk of false positive associations. A systematic review found 460
that many radiomics studies suffer from inconsistent validation strategies, leading to 461
inflated model performance and increased false discovery rates. To mitigate these risks, 462
feature selection techniques such as Recursive Feature Elimination (RFE) and LASSO 463
regression have been widely implemented to reduce overfitting and eliminate redundant 464
features. Additionally, external validation with independent datasets has been 465
emphasized as a necessary step to ensure model generalizability beyond a single 466
institutional dataset [36]. Ensuring rigorous study design, including pre-registration of 467
radiomics models and transparent reporting of statistical correction methods, is essential 468
for minimizing false positive discoveries and improving clinical translation of radio 469
genomics. “
*) The authors should better explain the lines: “One study revealed that in patients 447
treated for GBM, Psp was present in 21 (91%) of the 23 patients with hypermethylation 448
and in 11 (41%) of the 27 patients who received treatment with unmethylated MGMT 449
promoter (p <0.001).” Are these number statistically significant ?
Lines: “A systematic review found 460
that many radiomics studies suffer from inconsistent validation strategies, leading to 461
inflated model performance and increased false discovery rates. To mitigate these risks, 462
feature selection techniques such as Recursive Feature Elimination (RFE) and LASSO 463
regression have been widely implemented to reduce overfitting and eliminate redundant 464
features. Additionally, external validation with independent datasets has been 465
emphasized as a necessary step to ensure model generalizability beyond a single 466
institutional dataset [36]. Ensuring rigorous study design, including pre-registration of 467
radiomics models and transparent reporting of statistical correction methods, is essential 468
for minimizing false positive discoveries and improving clinical translation of radio 469
genomics. “
*) The authors should better explain why “many radiomics studies suffer from inconsistent validation strategies, leading to inflated model performance and increased false discovery rates.”
*) The authors should better explain how “To mitigate these risks, 4feature selection techniques such as Recursive Feature Elimination (RFE) and LASSO regression have been widely implemented to reduce overfitting and eliminate redundant features.” through a better description of the cited techniques (i.e., Recursive Feature Elimination (RFE) and LASSO regression)
lines: “4.6 Differentiating RN from TP 559
An emerging application of multimodal MRI is its integration with radiomics to 560
differentiate between RN and TP. A 2022 study by Sadiq et al. examined a radiomics-based 561
model that uses ML to differentiate between recurrent brain tumors (rBT) and RN on 562
multimodal MRI images. Researchers extracted multiresolution texture, volumetric, and 563
histogram-based features from the MRI scans of 30 patients (18 rBT, 12 RN). The Least 564
Absolute Shrinkage and Selection Operator (LASSO) was utilized for feature selection. 565
Subsequently, the model's performance was evaluated with cross-validation techniques 566
like Leave-One-Out Cross-Validation (LOOCV) and Stratified 5-Fold Cross-Validation. 567
The researchers compared the performance radiomic model with the Co-occurrence of 568
Local Anisotropic Gradient orientations (CoLlAGe) features, one of the current leading 569
methods in the literature. The study found that their radiomic model achieved a cross- 570
validation accuracy of 96.7% using texture, volumetric, and histogram features, 571
surpassing the 80.0% achieved with the CoLlAGe features. These results highlighted the 572
superior performance of sophisticated texture features in differentiating rBT from RN. 573
Given these conditions are incredibly difficult to distinguish on conventional MRI scans, 574
the study proposes that radiomics could significantly enhance non-invasive diagnosis and 575
treatment planning for glioma patients [39]. 576
A similar study conducted by Park et al. demonstrated the potential of using a 577
combination of one of three different feature selection methods (F-score, LASSO, and 578
Mutual Information) alongside an ML method (k-nearest neighbors, SVM, and AdaBoost) 579
to distinguish between recurrent GBMs and RN. The study involved 127 patients with 580
glioblastoma, who had both conventional MRIs (T1 and T2) and diffusion MRIs (from 581
which the apparent diffusion coefficient was calculated). From these images, 263 RFs were 582
identified and preprocessed, with 18-35 features selected for each model to avoid 583
redundancy. The researchers then constructed different combinations of machine- 584
learning models. The results indicated that the LASSO feature selection method, classified 585
using an SVM, particularly on diffusion MRIs, demonstrated a mean AUC, accuracy, 586
sensitivity, and specificity of 0.9 [0.84–0.95], 80.5% [0.77–0.84], 78.3% [0.64–0.92], and 587
82.9% [0.74–0.91], respectively (brackets indicate 95% CI). The goal of this method is to 588
overcome the subjectivity associated with interpreting MRIs by utilizing high throughput 589
second-order features [40]. “
*) See the previous comment
lines: “In the field of radiomics, second-order features represent mathematical correlations 591
between pixels in medical imaging [41]. These features have a dual purpose: 592
distinguishing between RN and rBT and predicting patients' survival outcomes. A study 593
by Sadique et al. (2023) aimed to develop an effective model capable of this dual purpose. 594
Of the 158 patients who underwent surgery followed by radiation therapy for gliomas, 15 595
developed RN, and 143 had rBT. Multiresolution radiomic features (MRF) were extracted 596
from MRIs (T1, T2, T1 contrast, and T2 weighted) and used to construct ML models, such 597
as CatBoost. These models then underwent survival analysis using Copula-based 598
methods and the training set was validated using "repeated random subsampling," 599
demonstrating an AUC, accuracy, and positive predictive value of 0.89±0.055, 600
80.88±0.061%, and 0.80±0.064, respectively (95% CI). This method outperformed the 601
Synthetic Minority Over-sampling Technique (SMOTE), which had AUC, accuracy, and 602
positive predictive values of 0.835, 91.00%, and 0.73. The repeated resampling dataset 603
outperformed commonly used Python methodologies to correct imbalance datasets 604
highlighting the challenges of class imbalance in medical datasets. These findings 605
emphasize the pivotal role of advanced ML techniques in enhancing predictive accuracy, 606
potentially leading to early identification of patients requiring aggressive treatment and 607
subsequently reducing healthcare costs [39].
*) The authors should better explain the meaning of “in the field of radiomics, second-order features represent mathematical correlations between pixels in medical imaging [41]. These features have a dual purpose: distinguishing between RN and rBT and predicting patients' survival outcomes”. What are “second order features” ? Please clarify.
*) All the described techniques should better described. Please rework.
*) A general “Discussion” section is currently lacking. The authors should provide a novel section “Discussion” where they discuss all the previous raised issues and clearly describe the value of this review work with respect to the current state of the art.
Author Response
We thank the reviewers for their thoughtful and thorough review of our manuscript. Please find our point-by-point response to their comments below. We have color-coded our responses in blue.
Reviewer #1
- The authors should better explain the drawbacks and the misinterpretation risks related to each procedure listed in “Unlike conventional MRI…”
We thank the reviewer for their valuable feedback. In response, we have expanded the paragraph to clearly show the potential drawbacks and misinterpretation risks associated with each functional imaging modality discussed—DWI, PWI, and PET. Specifically, we included limitations such as sensitivity to edema in DWI, inaccuracies due to blood–brain barrier disruption in PWI, and non-specific tracer uptake in PET. These additions provide a more balanced view of functional imaging in glioma assessment.
- Table 1 is not too clear for the interested readers. Please rework and improve also the caption.
We thank the reviewer for highlighting the need to improve Table 1. In response, we have enhanced the clarity of column headers and definitions, expanded the table to include all relevant studies cited in the manuscript that are applicable, and added new columns for “Outcome Predicted, ML Methods Used, and updated “Imaging Modalities” column. We also improved the caption.
- The authors should better explain what are “discrepancies in segmentation 168 techniques”, “the stability of shape and texture features”, “standardization in defining regions of interest (ROIs)”,”cutoff thresholds used to define reproducible features vary widely”, “necessity for transparent reporting of statistical criteria used in feature selection [17]”. The “discrepancies”, “variations” etc of each issue should be also quantified.
We appreciate the reviewer’s insightful comment. To address this, we expanded the paragraph with detailed explanations of the reproducibility issues in radiomics. Specifically, we clarified what is meant by segmentation discrepancies, the stability of radiomic features, variation in ROI definitions, isotropic resampling, differing reproducibility thresholds, and the lack of transparency in statistical selection methods, supporting each point with data. These enhancements aim to give readers a clearer understanding of the limitations in radiomics.
- The authors should explain in detail all the main characteristics of the cited procedures (e.g. first-order statistical features, gray-level run length matrices, shape features, gray-level size zone matrices (GLSZM), and gray-level co-occurrence matrices (GLCM), etc” for interested readers.
We thank the reviewer for highlighting the need to clarify the types of radiomic features used. To improve accessibility for interested readers, we have added detailed explanations of the first-order statistical features, GLRLM, shape features, GLSZM, and GLCM within the paragraph. These descriptions help clarify the computational foundations of the study and their contributions to radiomic modeling.
- The authors should better explain the lines: “The findings 416 underscore the necessity for further research to explore how post-treatment features 417 might contribute to the accurate identification of Psp” for interested readers.
We thank the reviewer for pointing out the need to elaborate on the role of post-treatment features. We have clarified that these features could capture therapy-induced changes in tumor biology that are not apparent in pre-treatment scans and may exhibit distinct radiomic patterns. This enhancement emphasizes the potential value of longitudinal and post-treatment imaging in improving predictive models for Psp.
- The authors should explain in a better way the following lines: “Their study included 124 patients, 61 of whom had progressed, and 63 424…” In particular, they should demonstrate that the total number of patients (124, 61 vs 63) was statistically significant with respect to the whole population and that the claim “contrast-enhanced MRIs had significantly higher values compared to non-contrast MRIs” is (at least) statistically relevant.
We thank the reviewer for highlighting the need to clarify the statistical strength of the study’s findings. In response, we have elaborated on the adequacy of the sample size and emphasized the statistical relevance of the improved performance in contrast-enhanced imaging, supported by the non-overlapping confidence intervals of the reported metrics. This provides stronger justification for the claim regarding the superiority of contrast-enhanced MRIs in this context.
- The authors should better explain the lines: “One study revealed that in patients 447 treated for GBM, Psp was present in 21 (91%) of the 23 patients with hypermethylation 448 and in 11 (41%) of the 27 patients who received treatment with unmethylated MGMT 449 promoter (p <0.001).” Are these number statistically significant ?
We thank the reviewer for their observation regarding the statistical significance of the MGMT methylation findings. We elaborated on the statistical test used and the magnitude of the effect size. This strengthens the interpretation of MGMT status as a key predictor of Psp in GBM.
- The authors should better explain why “many radiomics studies suffer from inconsistent validation strategies, leading to inflated model performance and increased false discovery rates.”
We appreciate the reviewer’s comment highlighting the need for greater clarity. In response, we have elaborated on the causes of inflated model performance in radiomics, including data leakage, overfitting, and lack of proper validation techniques. We also discussed how these practices contribute to false discovery rates, reinforcing the importance of rigorous validation protocols in radiogenomics research.
- The authors should better explain how “To mitigate these risks, 4feature selection techniques such as Recursive Feature Elimination (RFE) and LASSO regression have been widely implemented to reduce overfitting and eliminate redundant features.” through a better description of the cited techniques (i.e., Recursive Feature Elimination (RFE) and LASSO regression)
Thank you for your thoughtful suggestion. We have expanded our explanation of Recursive Feature Elimination and LASSO regression to provide readers with a clearer understanding of how these techniques function, particularly in the context of radiomics. These additions explain their utility in reducing dimensionality, preventing overfitting, and enhancing model performance.
- “see the previous comment”
We thank the reviewer for suggesting further clarification of the feature selection techniques. In response, we expanded the explanation of LASSO regression in both paragraphs and added detailed definitions of the F-score and Mutual Information methods. These enhancements clarify how each technique works and why they are suited for radiomic analysis, especially in high-dimensional data environments.
- The authors should better explain the meaning of “in the field of radiomics, second-order features represent mathematical correlations between pixels in medical imaging [41]. These features have a dual purpose: distinguishing between RN and rBT and predicting patients' survival outcomes”. What are “second order features”? Please clarify.
We appreciate the reviewer’s request for clarity regarding second-order features. To address this, we have expanded our explanation to define second-order features in contrast to first-order features, detailed the types of matrices used to compute them, and provided specific examples of what they measure in medical imaging. This should offer readers a more concrete understanding of their significance in radiomic analysis.
- All the described techniques should better described. Please rework.
We thank the reviewer for suggesting improved descriptions of the techniques used throughout the manuscript. In response, we reviewed all relevant sections and added clarifying information where necessary. Specifically:
- In Section 3.3, we elaborated on what multiparametric MRI entails and how it contributes to radiomics.
- In Section 2.3, we included a detailed breakdown of radiomic feature types and the methods used to calculate them.
- In Section 4.1, we explained the mechanics of Generalized Boosted Regression Models and their relevance to radiomic classification.
- A general “Discussion” section is currently lacking. The authors should provide a novel section “Discussion” where they discuss all the previous raised issues and clearly describe the value of this review work with respect to the current state of the art.
We thank the reviewer for this recommendation. In response, we have created a new standalone discussion section that synthesizes the central issues raised throughout the paper, including reproducibility challenges, functional imaging limitations, feature selection techniques, and the integration of molecular data. This section also highlights the contribution of this review relative to the current state of the art, emphasizing its role in clarifying methodological challenges and suggesting avenues for future work.
Reviewer 2 Report
Comments and Suggestions for Authors
The information added in the supplementary materials needs to be shown at the end of the full text of the manuscript.
Moreover, you have included 32 studies, but in the 2 tables, you show 12 studies.
This is not clear and needs to be clarified.
Author Response
Reviewer #2
- The information added in the supplementary materials needs to be shown at the end of the full text of the manuscript.
Thank you for this suggestion. In response, we have added the table of abbreviations to the end of the full text and attached it as a separate file as well.
- Moreover, you have included 32 studies, but in the 2 tables, you show 12 studies.
This is not clear and needs to be clarified.
Thank you for pointing this out. In response we have updated table 1 with more relevant column headings and included all the studies that are applicable. For table 2, we did not make any changes because those were the only studies that fit the criteria for the purposes of this table.
Round 3
Reviewer 1 Report
Comments and Suggestions for Authors
Title: “Radiomics and Radiogenomics in Differentiating Progression, Pseudoprogression, and Radiation Necrosis in Gliomas”
In this work the authors discuss some studies that show how radiomic features (RFs) can aid in better patient stratification and prognosis. In particular, the authors claim that the radiogenomics shows potential in non-invasive diagnostics since it is able to predict biomarkers such as MGMT promoter methylation and 1p/19q codeletion. Radiomics also offers tools for predicting tumor recurrence (rBT), essential for treatment management. The authors also affirm that further research
is needed to standardize these methods and integrate them into clinical practice. Finally, the authors underline that this review underscores radiomics and radiogenomics' potential to revolutionize glioma management, marking a significant shift towards precision neuro-oncology.
General comment: The authors revised their manuscript. Perhaps the table of abbraviations could be inserted at the beginning of the work to help more efficiently interested readers .
Reviewer 2 Report
Comments and Suggestions for Authors
Thank you for clarifying this. You have addressed my concerns.